# Cross-Sectional Survey of Antibiotic Resistance in Extended Spectrum β-Lactamase-Producing *Enterobacteriaceae* Isolated from Pigs in Greece

**DOI:** 10.3390/ani12121560

**Published:** 2022-06-16

**Authors:** Nikolaos Tsekouras, Zoi Athanasakopoulou, Celia Diezel, Polychronis Kostoulas, Sascha D. Braun, Marina Sofia, Stefan Monecke, Ralf Ehricht, Dimitris C. Chatzopoulos, Dominik Gary, Domenique Krähmer, Vassiliki Spyrou, Georgios Christodoulopoulos, Charalambos Billinis, Vasileios G. Papatsiros

**Affiliations:** 1Clinic of Medicine, Faculty of Veterinary Science, University of Thessaly, 43100 Karditsa, Greece; nitsekou@vet.uth.gr (N.T.); gc@vet.uth.gr (G.C.); 2Department of Microbiology and Parasitology, Faculty of Veterinary Science, University of Thessaly, 43100 Karditsa, Greece; zathanas@uth.gr (Z.A.); msofia@uth.gr (M.S.); billinis@uth.gr (C.B.); 3Leibniz Institute of Photonic Technology (IPHT), 07745 Jena, Germany; celia.diezel@leibniz-ipht.de (C.D.); sascha.braun@leibniz-ipht.de (S.D.B.); stefan.monecke@leibniz-ipht.de (S.M.); ralf.ehricht@leibniz-ipht.de (R.E.); 4InfectoGnostics Research Campus, 07745 Jena, Germany; 5Faculty of Public and One Health, University of Thessaly, 43100 Karditsa, Greece; pkost@uth.gr (P.K.); dchatzopoulos@uth.gr (D.C.C.); 6INTER-ARRAY by fzmb GmbH, 99947 Bad Langensalza, Germany; dgary@fzmb.de (D.G.); dkraehmer@fzmb.de (D.K.); 7Faculty of Animal Science, University of Thessaly, 41110 Larissa, Greece; vasilikispyrou@uth.gr

**Keywords:** antimicrobial resistance, pigs, ESBL-producing *Enterobacteriaceae*, multidrug resistance genes, Greece

## Abstract

**Simple Summary:**

The objective of this study was to investigate, for the first time in Greece, the prevalence of ESBL producers in swine populations and to correlate their occurrence with risk factors. A total of 214 fecal samples were collected from the farms from December 2019 to April 2021. A subset of 78 (78/214, 36.5%) ESBL producers were identified as *Escherichia coli* (*E. coli*, 88.5%), *Klebsiella pneumoniae* spp. *pneumoniae* (*K. pneumoniae*, 3.8%), *Proteus mirabilis* (*P. mirabilis*, 5.1%), *Enterobacter cloacae* complex (*E. cloacae* complex, 1.3%) and *Salmonella enterica* spp. *diarizonae* (*S. enterica* spp. *diarizonae*, 1.3%). CTX-M, SHV and TEM genes were detected along with genes conferring resistance to fluoroquinolones, aminoglycosides, sulfonamides, trimethoprim, macrolides and colistin. This study displayed high antimicrobial resistance rates in the Greek swine industry, and our results are alarming for both human and animal health.

**Abstract:**

This study aimed to estimate the prevalence of extended-spectrum β-lactamase-producing (ESBL) bacteria in swine. Thus, 214 fecal samples were collected from suckling and weaned piglets from 34 farms in Greece (out of an overall population of about 14,300 sows). A subset of 78 (36.5%) ESBL producers were identified as *E. coli* (69/78, 88.5%), *K. pneumoniae* spp. *pneumoniae* (3.8%), *P. mirabilis* (5.1%), *E. cloacae* complex (1.3%) and *S. enterica* spp. *diarizonae* (1.3%). Resistance to at least one class of non-β-lactam antibiotics was detected in 78 isolates. Among the *E. coli* strains, resistance was identified with regard to aminoglycosides (*n* = 31), fluoroquinolones (*n* = 49), tetracycline (*n* = 26) and trimethoprim/sulfamethoxazole (*n* = 46). Of the three *K. pneumoniae* spp. pneumoniae, two displayed resistances to aminoglycosides and all were resistant to fluoroquinolones, tetracyclines and trimethoprim/sulfamethoxazole. As for the four *P. mirabilis* isolates, three had a resistant phenotype for aminoglycosides and all were resistant to imipenem, fluoroquinolones, tetracyclines and trimethoprim/sulfamethoxazole. Molecular characterization of the isolates revealed the presence of CTX-M, SHV and TEM genes, as well as of genes conferring resistance to fluoroquinolones, aminoglycosides, sulfonamides, trimethoprim, macrolides and colistin. High levels of antimicrobial resistance (AMR) were demonstrated in Greek swine herds posing a concern for the efficacy of treatments at the farm level as well as for public health.

## 1. Introduction

By the 1950s, antibiotics were being regularly used in industrial livestock production to secure animals’ health and improve their productivity [1]. Antibiotics are typically used in farms, not only for treatment purposes, but also for regulating the spread of infections (metaphylaxis), inhibiting infections (prophylaxis), especially in high-stress periods (weaning stage, post-vaccination period, after farrowing), and improving feed intake and growth performance [2]. Subtherapeutic doses of antimicrobials were administered to livestock for decades to prevent diseases and/or to enhance growth [3,4], a strategy that promoted the development and spread of antimicrobial resistant strains. Nowadays, the use of antibiotics (e.g., amoxicillin) is included at routine metaphylaxis programs. For instance, injectable amoxicillin can be administered postpartum for metaphylaxis of postpartum dysgalactia syndrome (PPDS) or in weaning feed for the prevention and metaphylaxis of post-weaning diarrhea (PWD) [5,6].

Antimicrobial resistance (AMR) in food-producing animals has drawn global attention, as approximately 70% of the overall antibiotic consumption in Europe is related to the animal sector [7,8]. A major outcome of AMR dissemination is non-effective treatments of livestock, which are further associated with decreased productivity and economic losses due to increased treatment costs [9]. Considering the estimated continuous rise in the global demand for animal derived products, consumption of antimicrobials by livestock is anticipated to increase by two-thirds over the next years [10]. In fact, antimicrobial consumption is estimated to be greater in pigs in comparison to chicken and cattle production systems [10] and a greater possibility of AMR has already been reported in pigs than in chicken and other food animals, or aquaculture [11,12,13]. Furthermore, AMR has been demonstrated in wild boars, a wildlife species that can act as a source of zoonotic pathogens causing human diseases, such as colibacillosis, salmonellosis, yersiniosis and listeriosis [14,15]. AMR in domestic and wild animals poses a hazard for human health by introducing resistant pathogens into the food chain and by triggering horizontal transfer of resistance determinants to other bacteria [16].

Extended spectrum β-lactamase-producing bacteria (ESBL) display resistance to the commonly used beta-lactam antimicrobial agents, including third generation cephalosporins, such as ceftriaxone, ceftazidime and ceftiofur [17]. β-lactams, namely penicillins, carbapenems, monobactams and cephalosporins, constitute 60% (by weight) of all antibiotics used worldwide, and are among the most extensively prescribed antibiotic classes in human medicine [18,19]. Due to excessive usage of β-lactams in both humans and animals, an increased spread of ESBL-producing bacteria has been observed, threatening personnel in the swine industry, and consequently posing a threat to human health [20]. ESBLs are widespread in *Enterobacteriaceae*, especially in *E. coli* and *Salmonella* spp., while ESBL-producing *E. coli* has been reported in food animals worldwide [21,22]. Previous reports have associated human ESBL carriage with exposure to ESBL-producing *Enterobacteriaceae* of livestock origin, raising concerns about the possible transfer of ESBL producers through the food chain, which could jeopardize public health [23,24,25,26,27,28].

Reviewing previous literature revealed the scarcity of published data about ESBL-producing *Enterobacteriaceae* from pig farms in Greece. Thus, the objective of this study was to report, for the first time in Greece, the prevalence of ESBL producers in pig herds, to phenotypically and molecularly identify their antimicrobial resistance patterns and to investigate potential factors that could promote the development of AMR.

## 2. Materials and Methods

### 2.1. Ethics

All procedures were performed according to the ethical standards in the Helsinki Declaration of 1975, as revised in 2000, as well as the national law, and after receiving approval (number 96/19.12.2019) from the Institutional Animal Use Ethics Committee of the Faculty of Veterinary Science, University of Thessaly.

### 2.2. Study Design

The current cross-sectional study was conducted in different regions of Greece for two years (between 2019–2021) and included 34 pig farms. The farms had an overall population of about 14,300 sows, which represented approximately 24% of the entire capacity of the Greek swine production. Farms were in northern (*n* = 4), central (*n* = 13), western (*n* = 10) and southern (*n* = 7) Greece. Generally, central and western Greece are the regions with the highest pig density (more than 50% of total pig population of Greece). The classifications of pig farms according to their geographic origin and capacity are presented in Table 1.

Farmers or managers consented to participating in the study. The inclusion and exclusion criteria for farm selection are shown in Table 2. These criteria were met by all participating herds in order to include intensive farms that employ common practices of the Greek swine production system, while taking preventive measures for disease control.

In addition, data concerning sow age and prior administration of antimicrobials for the last six months were collected. The most used classes of antibiotics were penicillins, collistin, cephalosporins, quinolones, pleuromoutilins (PLMs), macrolides, tetracyclines and trimethoprim/sulfonamides; these were used according to the manufacturers’ instructions with respect to duration of therapy and dosage.

### 2.3. Sample Collection

A total of 214 fecal samples were collected from 73 suckling and 141 weaning piglets of 34 farms (Table 1). Samples were obtained directly from the rectum by using swabs with Amies transport medium (Transwab^®^, Amies, UK) and were transferred within a day to the Laboratory of Microbiology and Parasitology (Faculty of Veterinary Medicine, Karditsa, Greece).

### 2.4. Isolation and Identification of Extended Spectrum Cephalosporin Resistant (ESCR) Strains

For the detection of ESCR isolates, fecal swabs were directly streaked on ESBL selective media (CHROMID^®^ ESBL, BioMérieux, Marcy l’Etoile, France) and the plates were incubated aerobically at 37 °C for 24–48 h. Subcultures were grown on both MacConkey agar and 5% sheep blood agar until pure cultures were obtained. Bacterial species identification was carried out using the automated Vitek-2 system (BioMérieux. Marcy l’Etoile, France), according to the manufacturer’s instructions.

### 2.5. Isolation and Identification of Salmonella spp.

Isolation of *Salmonella* spp. was conducted, according to ISO 6579-1:2017. Initially, swabs were agitated and squeezed into sterilized tubes containing 9 mL Buffered Peptone Water (BPW). Subsequently, Modified Semisolid Rappaport-Vassiliadis (MSRV) agar, Xylose Lysine Deoxycholate (XLD) agar, and *Salmonella Shigella* (SS) agar were used as selective media under the recommended conditions. All presumptive *Salmonella* colonies were identified as to species using the Vitek-2 system.

### 2.6. Antimicrobial Susceptibility Testing

Antimicrobial susceptibility testing of all the obtained strains was performed by the Vitek-2 system. The AST-GN96 card was used to determine the minimum inhibitory concentration (MIC) of the following antimicrobial classes: penicillins (ampicillin-AMP, amoxicillin/clavulanic acid-AMC, ticarcillin/clavulanic acid-TCC), cephalosporins (cefalexin-CEX, cefalotin-CF, cefoperazone-CEP, ceftiofur-CEF, cefquinome-CEQ), carbapenems (imipenem-IMI), aminoglycosides (gentamicin-GEN, neomycin-NEO), quinolones (flumequine-FLU, enrofloxacin-ENR, marbofloxacin-MRX), tetracyclines (tetracycline-TET), amphenicols (florfenicol-FLO), polymyxin B-PL and sulfonamides (trimethoprim/sulfamethoxazole-SXT).

### 2.7. Phenotypic Confirmation of ESBL Production

All the isolates that presented resistance to 3rd (CEP, CEF)-generation cephalosporins were screened via the double disk synergy test (DDST) or a combination disk test (CDT) for ESBL production, according to the European Committee on Antimicrobial Susceptibility Testing (EUCAST) guidelines [30]. In brief, antibiotic disks containing cefotaxime (CTX) (30 μg), ceftazidime (CAZ) (30 μg), cefepime (CPM) (30 μg) and AMC (20 μg/10 μg) were applied at a distance of 20 mm (center to center) on Mueller Hinton agar previously inoculated with an 0.5 McFarland inoculum of the isolate to be tested. After incubation, any enhanced zone of inhibition between cephalosporin disks and the AMC disk or a ‘’keyhole’’ formation in the direction of the disk containing clavulanic acid were considered as evidence for the presence of an ESBL-producing strain. In cases of ambiguous results, a combination disk test was also applied, using CTX and CAZ disks (30 µg each), alone and in combination with clavulanic acid (10 µg). A difference of ≥5 mm in zone diameter between the test using the disks alone and that using the disks combined with clavulanic acid antimicrobial agents was interpreted as ESBL production.

### 2.8. Antibacterial Resistance Genes of ESBL-Producing Enterobacteriaceae

Isolates that were found to be positive in the DDST or the CDT were characterized using the DNA microarray-based assay CarbaResist from InterArray (FZMB GmbH, Bad Langensalza, Germany). Primer and probe sequences have previously been described in detail [31]. In addition, probes for the detection of the colistin resistance gene family mcr were included on the present microarray (see Appendix A). Protocols and procedures were conducted in accordance with the manufacturer’s instructions (https://www.inter-array.com/Further-Genotyping-Kits, accessed on 10 May 2022). In brief, bacteria were grown overnight on Columbia blood agar. Bacteria were harvested and genomic DNA was extracted using the Qiagen blood and tissue kit (Qiagen, Hilden, Germany), following the manufacturer’s instructions. The DNA was used in a multiplexed primer elongation incorporating biotin-16-dUTP. Amplicons were stringently hybridized to the microarray, washed and incubated with a horseradish-peroxidase-streptavidin conjugate. Hybridizations were detected by adding a precipitating dye.

### 2.9. Statistical Analysis

Principal component analysis (PCA) and hierarchical clustering were used to explore the patterns of antimicrobial co-resistance among the isolated *Enterobacteriaceae* species and identify clusters of co-resistance [32]. Subsequently, the prevalence of AMR by species was estimated within a Bayesian estimation framework [33].

Logistic regression models were used to assess whether (a) the presence of ESBL-producing strains and (b) antimicrobial resistance to a certain type of antibiotic are associated with a series of candidate variables. Candidate variables for both (a) and (b) were herd size, sow age and administration of antibiotics (penicillins, collistin, cephalosporins, quinolones, pleuromoutilins (PLMs), macrolides, tetracyclines and trimethoprim/sulfonamides) according to the manufacturers’ instructions. All candidate variables were initially screened, one-by-one, with a significance level of 0.25. For the shortlisted variables, collinearity analyses were conducted to identify pairs of collinear variables. For each pair of collinear variables, one was excluded from further analyses. The variable that was retained was the one more strongly associated with the outcome. Variables with *p* < 0.25 were then tested in the final model and were subsequently reduced by backwards elimination, until only significant (*p* < 0.05) variables remained.

All analyses were performed in R program [34]. For PCA, we used the prcomp built-in functions, and for Bayesian prevalence estimation, the runjags package [35] and figures were built with the ggplot2 package [36]. For logistic regression, the glmer function was used [37].

## 3. Results

### 3.1. Isolation and Identification of ESBL-Producing Enterobacteriaceae

A total of 98 ESCR strains were recovered by selective cultivation from 95 of the 214 (44.4%) swine samples tested. Additionally, five *Salmonella* spp. isolates were retrieved from an equal number of samples (2.3%).

Seventy-eight (36.5%) isolates presented resistance to 3rd generation cephalosporins and were phenotypically confirmed to produce ESBL. ESBL producers were identified as *E. coli* (*n* = 69), *K. pneumoniae spp. pneumoniae* (*n* = 3), *P. mirabilis* (*n* = 4), *E. cloacae* complex (*n* = 1) and *S. enterica subsp. diarizonae* (*n* = 1). The results are summarized in Table 3.

### 3.2. Antimicrobial Resistance Phenotype and Genotype of the ESBL-Producing Enterobacteriaceae

All ESBL isolates (*n* = 78) presented resistance to AMP and to all the cephalosporins tested, apart from four *E. coli* strains, which were susceptible to CEQ, and the isolate of the *E. cloacae* complex, which was susceptible to CEP. Resistances to AMC (33.3%) and TCC (26.9%) were also detected, while diminished susceptibility to imipenem was identified only in the four *P. mirabilis* isolates (5.1%). ESBL phenotypes are illustrated as percentages in Figure 1. Three *E. coli* isolates could not be retrieved after storage in −80 °C, and were thus not genotypically characterized. Of the remaining 66 *E. coli* isolates, ESBL genes were detected in 65. In particular, *bla*_CTX-M1/15_ was detected in 52 isolates (78.8%), *bla*_CTX-M9_ in six (9.1%), *bla*_CTX-M8_ in six (9.1%), *bla*_SHV_ in three (4.5%) and *bla*_TEM_ in 38 (57.6%), alone (*n* = 5) or in combination with other variants (*n* = 33). The *K. pneumoniae* isolates were found to harbor *bla*_CTX-M1/15_ (*n* = 2), *bla*_SHV_ (*n* = 2), *bla*_TEM_ (*n* = 1) and *bla*_CTX-M9_ (*n* = 1), while the *P. mirabilis* harbored *bla*_CTX-M9_ (*n* = 3), *bla*_CTX-M8_ (*n* = 1) and *bla*_TEM_ (*n* = 3). Finally, the *S. enterica* harbored *bla*_TEM_, whereas no ESBL genes were detected in the *E. cloacae*.

The AmpC gene *bla*_ACT_ was detected in 18 *E. coli* (27.3%) and the *bla*_CMY_ in one. Moreover, 21 isolates (31.8%) possessed *bla*_OXA-1_ and one *bla*_OXA-60_.

The detailed antimicrobial resistance phenotype and genotype of the ESBL isolates is reported in Appendix A.

### 3.3. Antimicrobial Resistance Phenotype and Genotype of ESBL-Producing Enterobacteriaceae to non β-lactam Antibiotics

All 78 ESBL-producing isolates displayed resistance to at least one class of non β-lactam antibiotics. Among the *E. coli* strains, resistances were reported for fluoroquinolones (FLU; *n* = 45, ENR; *n* = 42, MRX; *n* = 22), aminoglycosides (GEN; *n* = 19, NEO; *n* = 24), TET (*n* = 26) and SXT (*n* = 46) (Figure 1, Figure 2 and Figure 3).

Regarding *E. coli*, resistance genes were detected for fluoroquinolones in 35 isolates (*qnrA*, *qnrB*, *qnrS)*, for aminoglycosides in 59 (*aadA1*, *aadA2*, *aadA4*, *aphA*, *rmtA*, *rmtC*, *aac*(6′)-*Ib*, *aac*(3′)-*Iva*), for sulfonamides in 51 (*sul1*, *sul2, sul3)*, for trimethoprim in 58 (*dfrA1*, *dfrA12*, *dfrA13*, *dfrA14*, *dfrA15*, *dfrA17*, *dfrA19*, *dfrA5*, *dfrA7*), for macrolides in 16 (*mph*, *mrx*) and for colistin in seven (*mcr*-1/*mcr*-2, *mcr*-4, *mcr*-8). Additionally, the *intl1*, *intl2* and tnpISE*cp1* genes associated with mobile elements were identified in 20, seven and 30 *E. coli* isolates, respectively.

*K. pneumoniae* spp. *pneumoniae* isolates (*n* = 3) were resistant to fluoroquinolone (ENR; *n* = 3, FLU; *n* = 3, MRX; *n* = 1), TET and SXT, and two of the isolates displayed resistances to aminoglycosides (GEN; *n* = 2, NEO; *n* = 1) (Figure 1 and Figure 3). All three isolates harbored genes conferring resistance to fluoroquinolones (*qnrS*), sulfonamides (*sul1*, *sul2*) and trimethoprim (*dfrA14*, *dfrA7*, *dfrA17*, *dfrA1*), while two also possessed aminoglycoside-resistance genes (*aadA1*, *aphA*). Genes encoding mobile elements (*intl1*, *intl2*, tnpISE*cp1*) as well as the oqxAB multidrug efflux pump (*oqxA*, *oqxB*) were additionally detected in the three isolates.

*P. mirabilis* isolates (*n* = 4) were resistant to fluoroquinolones (ENR; *n* = 4, FLU; *n* = 3, MRX; *n* = 3), TET and SXT, while a resistant phenotype for aminoglycosides (GEN; *n* = 3, NEO; *n* = 1) was observed in three of them (Figure 1 and Figure 3). Antibacterial resistance genes revealed the presence of genes conferring resistance to sulfonamide (*sul1*, *sul2*) and trimethoprim (*dfrA1*, *dfrA5*, *dfrA17*) in all four of them, as well as of aminoglycoside-resistance genes in three (*aadA1*, *aadA2*, *aphA*, *aac*(6′)-*Ib*, *aac*(3′)-*Iva*). Furthermore, three of the strains harbored *intl2.*

The isolate of the *E. cloacae* complex presented intermediate resistance for NEO and TET, was resistant to FLU (Figure 1 and Figure 3) and was only found to harbor *dfrA5*.

Finally, the *S. enterica* spp. *diarizonae* exhibited resistance to fluoroquinolones (FLU, ENR), TET, SXT, aminoglycosides (GEN, NEO) and FLO (Figure 1 and Figure 3). The isolate carried resistance determinants against fluoroquinolones (*qnrS*), sulfonamides (*sul2*, *sul3*), trimethoprim (*dfrA12*, *dfrA17*) and aminoglycosides (*aadA1*, *aadA2*, *aadA4*, *aphA*), as well as the tnpISE*cp1*.

The antimicrobial resistance phenotype and genotype of the ESBL isolates are detailed in Appendix A. The resistance genes detected among the ESBL-producing Enterobacteriaceae are presented in Table 4.

### 3.4. Logistic Regression Analysis Results

Logistic regression analysis demonstrated that the presence of ESBL-producing *E. coli* strains was negatively associated with prior administration of PMLs, and that the occurrence of ESBL-producing *P. mirabilis* was associated with herd size. Further associations concerning ESBL-producing *Enterobacteriaceae* were not recognized by the model. The results are presented in Table 5.

The reported antimicrobial resistances of this study were not related to the sows’ ages. The development of resistances to AMP and AMC, as well as the diminished susceptibility to IMI, were positively associated with the farm size. By examining whether previous administration of antibiotics led to the development of AMR, we recognized a series of positive associations between (a) GEN resistance and previous administration of quinolones, (b) NEO resistance and previous usage of TETs, (c) MRX resistance and prior application of TETs and (d) ENR resistance and prior administration of cephalosporins. Finally, the previous application of PLMs was negatively associated with the development of resistance to both GEN and NEO. The results are described in Table 6.

## 4. Discussion

The present study aimed to describe, for the first time, the frequency of ESBL-producing *Enterobacteriaceae* from 34 pig farms located in different geographical regions of Greece, and to characterize their AMR phenotype and genotype. To that end, we collected and tested 214 fecal samples from 73 suckling and 141 weaning piglets from herds that met certain inclusion criteria. The most commonly isolated ESBL producers were *E. coli* strains that presented co-resistance to at least one class of non β-lactam antibiotics. In addition, *K. pneumoniae*, *P. mirabilis*, *E. cloacae* complex and *S. enterica* subsp. *diarizonae* isolates were identified as ESBL producers. Notably, four *P. mirabilis* strains displayed diminished susceptibility to IMI.

Resistance to AMC was observed in ESBL-producing *E. coli* isolates (Figure 3). In the studied farms, injectable amoxicillin was used postpartum in sows as part of a routine program of metaphylaxis for PPDS, and it was also added in weaning feed for the prevention and metalphylaxis of PWD. Resistance to amoxicillin has not become a major problem to date, because it is usually combined with clavulanic acid, a highly effective ESBL inhibitor [5]. As this antimicrobial is routinely administered in pigs against respiratory (e.g., bacterial pneumonia), enteric (post weaning diarrhea) and urogenital (e.g., PPDS) diseases, its unnecessary usage could further promote the selection of ESBL-producing bacteria. Therefore, laboratory diagnosis based on bacterial culture and sensitivity testing is mandatory prior to its application [5]. The observed AMC resistance of *E. coli* isolates retrieved from the piglets could be also associated with the administration of amoxicillin in sows postpartum, considering that vertical transmission of resistant bacteria from sows to piglets has been already documented [38]. However, future studies, including investigation of related risk factors (e.g., farm capacity and management, previous and current treatments with antibiotics, age, and treatment groups) will help us understand the development of AMR under field conditions.

ESBL-producing *E. coli* strains presented co-resistances to fluoroquinolones (*n* = 49), aminoglycosides (*n* = 31), TETs (*n* = 26) and SXT (*n* = 46), confirming previously reported antimicrobial resistance patterns of ESBL-producing *E. coli* [39,40,41,42]. In fact, the occurrence of strains resistant to AMP, AMC, SXT and TET [43,44], and the occurrence of ESBL producers resistant to at least one more class of non β-lactam antibiotics, have been formerly described in Greece [45]. We also observed increased resistance levels with regard to fluoroquinolones, which are widely used in swine clinical practice as first-choice agents for individual injectable treatment for respiratory, enteric and urogenital disease. This finding could be attributed to the extensive use of this agent at farm level and should be considered by veterinarians and farmers, especially in acute clinical cases that demand rapid treatment processes.

The occurrence of four *P. mirabilis* ESBL-producing isolates was reported and all strains presented reduced susceptibility to IMI. At first, this was an alarming finding, considering the prohibition of the use of IMI in livestock [46] and the reported carbapenemase resistance in humans [47,48]. As these isolates were not found to harbor a carbapenemase gene, the reported IMI resistance could be explained by pore mutations or mutations affecting a penicillin binding protein [49]. Furthermore, *P. mirabilis* isolates were also resistant to fluoroquinolones, TETs and SXT, while three of them had a resistant phenotype for aminoglycosides. This bacterial species is considered to be the most common etiological agent of PPDS in sows, causing severe economic losses in swine industry, whereas fluoroquinolones, TET and SXT are recommended as first-choice antimicrobial treatments [6]. Our results reveal a potential risk for treatment failure in field cases of PPDS, further emphasizing the importance of etiological diagnosis and sensitivity testing prior to the use of antibiotics by swine practitioners.

An ESBL-producing *S. enterica* subsp. *diarizonae* isolate was also noticed, presenting co-resistance to aminoglycosides, fluoroquinolones (FLU, ENR), TETs, SXT and FLO. Serovars of ESBL-producing *S. enterica* of poultry origin, co-resistant to aminoglycosides, TETs and SXT have previously been reported in Greece [50]. Herein, we identified a multidrug-resistant serovar in pigs that is usually isolated from humans [51], and thus, our results underline that AMR in livestock could pose, through the food chain, a serious threat to public health.

Molecular characterization of the ESBL isolates revealed the predominance of CTX-M type genes, mainly those of group 1, which is in accordance with preexisting literature about pigs [52,53,54]. Interestingly, various other genes conferring resistance to fluoroquinolones, aminoglycosides, sulfonamides, trimethoprim, macrolides and colistin were detected in the present study. This finding underlines the wide dissemination of AMR determinants among animals farmed for human consumption, which may have been facilitated by the detected mobile genetic elements [55]. The co-occurrence of *mcr* variants is especially noteworthy given the importance of colistin as a last-resort therapeutic option against multidrug-resistant strains in clinical settings. Colistin is commonly used in pigs to prevent and control the clinical outcomes of *E. coli* infection, including neonatal diarrhea, post-weaning diarrhea and edema disease [56]. Gene *mcr*-1, one of the most common colistin-resistance genes around the world, was detected among *E. coli* isolates from Chinese pigs at slaughter and retail meats, and its occurrence was speculated to be a result of colistin usage [57]. In Greece, multidrug-resistant *mcr*-1-positive ESBL-producing *E. coli* has previously been recovered on a dairy farm and was the etiological agent of mastitis [58].

Logistic regression analysis demonstrated a positive association between the herd size and the development of antimicrobial resistance to AMP, AMC and IMI, further confirming previous reports [59,60]. Specifically, it was more likely for AMR to appear in larger farms. A potential explanation for the observed AMP and AMC resistance could be provided by the routine administration of these antimicrobials, as infections in large herds are a more common phenomenon by reason of the higher swine density and the increased rates of gilt replacement from external sources [56]. On the contrary, diminished susceptibility to IMI cannot be ascribed to the usage of carbapenems, since this class of antibiotics is banned for swine due to the risk of compromising human treatments. In *P. mirabilis*, IMI resistance can be considered random, and it could be speculated that exposure to other beta-lactams might have been the driver (as the mechanism is unspecific).

At a national level, the usage of specific antimicrobials is strongly associated with the development of resistance towards these agents in commensal *E. coli* isolates in pigs, poultry and cattle [61]. We report a positive association between previous administration of TETs and the development of resistance to NEO and MRX, while increased GEN resistance was observed following quinolone administration. It is well-known that the use of different classes of antibiotics depends on the type of infections to be treated, and varies according to the pigs’ ages [1]. For instance, the gut microbiota is more exposed to orally administered tetracycline than to injected tetracycline, and thus an increase in AMR bacteria in the intestinal microbiome could be promoted by oral administration [62]. It is critical to expand our knowledge on the impact of the administration of different antibiotics in pigs, as increased resistance in bacteria may impair treatment efficacy and potentially lead to therapeutic failure.

Recent studies in Greece and worldwide have described the AMR profile of ESBL-producing *E. coli* strains isolated from livestock, including pigs [9,45,54]. Currently, as a wide dissemination of multidrug resistant bacteria in diverse ecosystems is observed, the establishment of an integrated antimicrobial surveillance system under the One Health approach is crucial for the protection of public health. The AMR profile of the ESBL-producing *Enterobacteriaceae* isolates presented in this study demonstrates the urgent need for the application of a monitoring program targeting preferentially multiple species.

## 5. Conclusions

We report a high percentage of ESBL-producing bacteria in the Greek swine industry, which is triggering alarm for veterinary practitioners and farmers as well as for public health authorities. The 78 ESBL producers (36.5%) were identified as *E. coli* (*n* = 69), *K. pneumoniae* (*n* = 3), *P. mirabilis* (*n* = 4), *E. cloacae* complex (*n* = 1) and *S. enterica* subsp. *diarizonae* (*n* = 1), and presented resistance to at least one class of non β-lactam antibiotics. CTX-M-1/15 enzymes were the most frequently observed ESBLs, followed by TEM and SHV types, and accompanied by several other resistance determinants for fluoroquinolones, aminoglycosides, sulfonamides, trimethoprim, macrolides, colistin and mobile genetic elements. As AMR increases the cost of animal production, antimicrobial usage must be prudent and based on laboratory diagnosis and antimicrobial sensitivity testing, in order to keep antibiotics as a therapeutic weapon in experts’ hands. The integration of a surveillance system for AMR monitoring under the One Health approach will contribute not only to the rational use of antibiotics, but also to the protection of public health.

## Figures and Tables

**Figure 1 animals-12-01560-f001:**
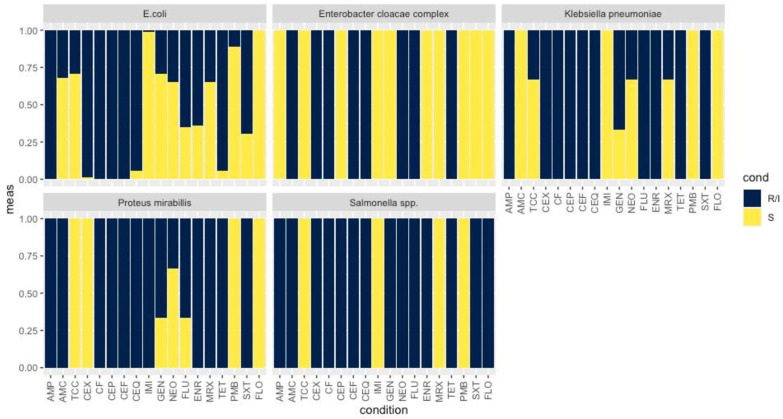
Bar plot showing the percentage of AMR of *E. coli*, *E. cloacae* complex, *K. pneumoniae*, *P. mirabilis* and *Salmonella* spp. to a range of antibiotics; AMP, AMC, TCC, CEX, CF, CEP, CEF, CEQ, IMI, GEN, NEO, FLU, ENR, MRX, TET, SXT and FLO. The values varied from 0.00 to 1.00; 0.00 indicates resistance and 1.00 susceptibility. R/I: Resistance/Intermediate results; S: Susceptibility; AMP: ampicillin, AMC: amoxicillin/clavulanic acid, TCC: ticarcillin/clavulanic acid, CEX: cefalexin, CF: cefalotin, CEP: cefoperazone, CEF: ceftiofur, CEQ: cefquinome, IMI: imipenem, GEN: gentamicin, NEO: neomycin, FLU: flumequine, ENR: enrofloxacin, MRX: marbofloxacin, TET: tetracycline, FLO: florfenicol, PMB: polymyxin B, SXT: trimethoprim/sulfamethoxazole (SXT).

**Figure 2 animals-12-01560-f002:**
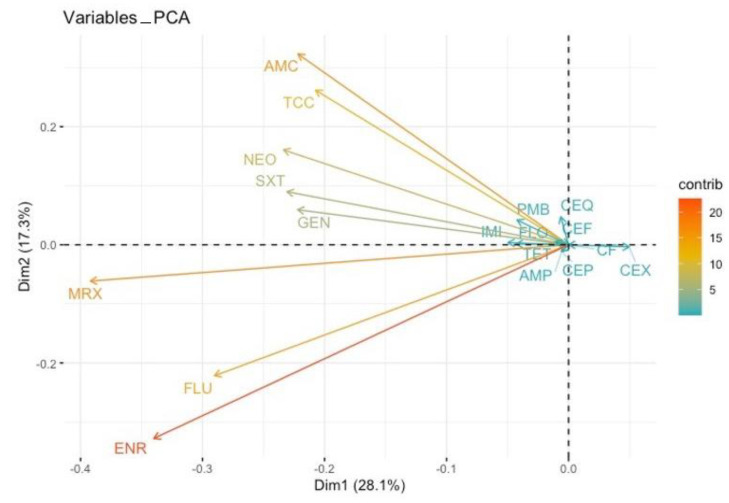
Principal component analysis (PCA) on co-resistances. Antibiotics are represented by vectors (arrows). Two vectors of antibiotics pointing in the same direction is an indication of a positive correlation between them; when we observe AMR in one antibiotic, we are expecting the development of AMR in the other one. An angle of 180 degrees between the vectors of two antibiotics is an indication of a negative correlation between them; when we observe AMR in one, we are not expecting the development of AMR in the other. A 90-degree angle between the vectors of two antibiotics indicates no relationship between them toward the developing AMR. The longer the vectors, the greater the intensity of this relationship. AMP: ampicillin, AMC: amoxicillin/clavulanic acid, TCC: ticarcillin/clavulanic acid, CEX: cefalexin, CF: cefalotin, CEP: cefoperazone, CEF: ceftiofur, CEQ: cefquinome, IMI: imipenem, GEN: gentamicin, NEO: neomycin, FLU: flumequine, ENR: enrofloxacin, MRX: marbofloxacin, TET: tetracycline, FLO: florfenicol, PMB: polymyxin B, SXT: trimethoprim/sulfamethoxazole (SXT).

**Figure 3 animals-12-01560-f003:**
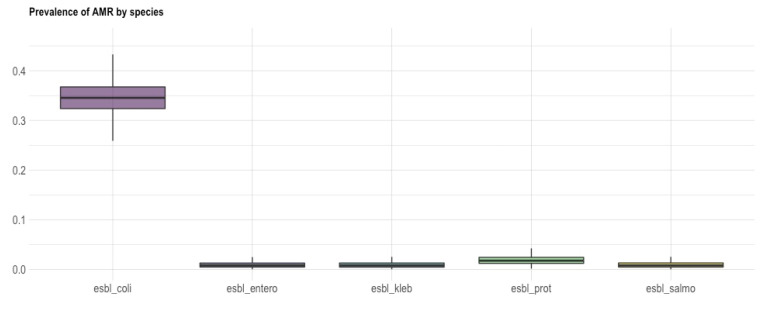
Box and whisker plot showing the prevalence of AMR (*y* axis) of ESBL-producing *E. coli*, *E. cloacae* complex, *K. pneumoniae*, *P. mirabilis* and *Salmonella* spp. (*x*-axis). The bold line shows the median value, while the length of the box represents the interquartile range.

**Table 1 animals-12-01560-t001:** Geographic origin and capacity of study’s farms.

**Region**	**Number of Farms**	**Total Capacity**	**Capacity per Farm**					**Number of Samples per Age**	
			**50–100**	**101–250**	**251–450**	**451–700**	**701–2100**	**Suckling Piglets**	**Weaning Piglets**
Northern Greece	4	1640	0	2	1	1	0	8	19
Central Greece	13	3710	4	2	3	4	0	24	44
Western Greece	10	5790	0	4	2	2	2	22	42
Southern Greece	7	3190	0	4	2	0	1	19	36
Total	34	14,300	4	12	8	7	3	73	141

**Table 2 animals-12-01560-t002:** Criteria for farms selection.

Criteria	Farms	Sows	Weaners
Capacity	Minimum 50 sows		
Type of farm	Exclusive farrow-to-finish		
Antiparasitic treatment		Antiparasitic treatment (IM) before farrowing	
Vaccination		Aujeszky’s disease virus Porcine parvovirus Atrophic rhinitis Erysipelas Porcine Reproductive and Respiratory Virus *E. coli* *Clostridium* spp.	Porcine Circovirus type 2 *Mycoplasma hyopneumoniae*
Diet		Home-made diets (mixed corn/barley/wheat–soybean-based meal) balanced in dietary nutrients (essential amino acids, minerals and vitamins) ^1^	
Toxin binders		Systematically used in the feed during gestation and lactation	Systematically used in the feed
Routine program for metaphylaxis of PPDS		Amoxicillin (IM) after farrowing	
Routine program of metaphylaxis for PWD			Amoxicillin via feed for the first 10 days after weaning

^1^ According to Nutrient Requirements of Swine (NRC) [29].

**Table 3 animals-12-01560-t003:** Isolation and identification of ESBL-producing bacteria from pig samples.

Bacterial Species	Percentage of Isolates (*n* = 214) ^1^	Percentage of ESBL-Producing Isolates (*n* = 78) ^2^
*Escherichia coli*	32.2 (*n* = 69)	88.5 (*n* = 69)
*Klebsiella pneumoniae* spp. *pneumoniae*	1.4 (*n* = 3)	3.8 (*n* = 3)
*Proteus mirabilis*	1.9 (*n* = 4)	5.1 (*n* = 4)
*Enterobacter cloacae*	0.5 (*n* = 1)	1.3 (*n* = 1)
*Salmonella enterica* subsp. *diarizonae*	0.5 (*n* = 1)	1.3 (*n* = 1)
Total	36.5 (*n* = 78)	100 (*n* = 78)

^1^ Number of fecal samples tested; ^2^ Number of isolated ESBL-producing bacteria.

**Table 4 animals-12-01560-t004:** Resistance genes detected among the ESBL-producing Enterobacteriaceae.

Resistance Genes	Number of Isolates				
	*E. coli*	*K. pneumoniae*	*P. mirabilis*	*S. enterica* ssp. *diarizonae*	*E. cloacae* Complex
*bla* _CTX-M1/15_	52	2	-	-	-
*bla* _CTX-M9_	6	1	3	-	-
*bla* _CTX-M8_	6	-	1	-	-
*bla* _SHV_	3	2	-	-	-
*bla* _TEM_	38	1	3	1	-
*bla* _ACT_	18	-	-	-	-
*bla* _CMY_	1	-	-	-	-
*bla* _OXA-1_	17	-	4	-	-
*bla* _OXA-60_	1	-	-	-	-
*aadA1*	36	1	3	1	-
*aadA2*	21	-	1	1	-
*aadA4*	14	-	-	1	-
*aphA*	21	1	1	1	-
*rmtA*	27	-	-	-	-
*rmtC*	7	-	-	-	-
*aac(6′)-Ib*	2	-	3	-	-
*aac(3′)-Iva*	1	-	3	-	-
*qnrS*	29	3	-	1	-
*qnrA*	2	-	-	-	-
*qnrB*	4	-	-	-	-
*sul1*	18	2	3	-	-
*sul2*	38	3	4	1	-
*sul3*	17	-	-	1	-
*dfrA1*	13	1	3	-	-
*dfrA5*	42	-	2	-	1
*dfrA7*	4	1	-	-	-
*dfrA12*	12	-	-	1	-
*dfrA13*	2	-	-	-	-
*dfrA14*	12	1	-	-	-
*dfrA15*	1	-	-	-	-
*dfrA17*	15	1	1	1	-
*dfrA19*	4	-	-	-	-
*mcr-1/mcr-2*	6	-	-	1	-
*mcr-4*	1	-	-	-	-
*mcr-8*	3	-	-	1	-
*mph*	12	-	-	-	-
*mrx*	16	-	-	-	-
*intl1*	20	2	-	-	-
*intl2*	7	1	3	-	-
*tnpISEcp1*	30	3	-	1	-
*oqxA*	1	3	-	-	-
*oqxB*	1	3	-	-	-

**Table 5 animals-12-01560-t005:** Associations of ESBL-producing Enterobacteriaceae with herd characteristics and administration of antibiotics.

ESBL-Producing	Logistic Regression			
	Parameter	Category	Estimate (95% CI)	*p* Value
*E. coli*	PMLs	0	1	
		1	0.25 (0.07; 0.84)	0.0215
*P. mirabilis*	Size	-	85.86 (4.02; 6620.06)	0.00648

**Table 6 animals-12-01560-t006:** Association of AMR with herd characteristics and administration of antibiotics.

Antibiotic	Logistic Regression			
	Parameter	Category	Estimate (95% CI)	*p* Value (Random Effect)
AMP	Size		48.72 (3.21; 1955.14)	0.0171
AMC	Size		31.38 (5.3; 290.81)	<0.005
IMI	Size		33.09 (2.16; 1454.48)	0.008
GEN	PMLs	0	1	
		1	0.15 (0.02; 0.58)	0.016
	Quinolones	0	1	
		1	13.14 (2.46; 243.71)	0.015
NEO	PMLs	0	1	
		1	0.19 (0.03; 1.05)	0.028
	TETs	0	1	
		1	7.7 (1.62; 59.44)	0.009
MRX	TETs	0	1	
		1	14.45 (0.96; 5.59)	0.011
ENR	Cephalosporins	0	1	
		1	3.49 (1.07; 14.84)	0.045

## Data Availability

All data generated for this study are presented within the manuscript and the Appendix A.

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
