# Peer review of "Cross-Sectional Survey of Antibiotic Resistance in Extended Spectrum β-Lactamase-Producing Enterobacteriaceae Isolated from Pigs in Greece"

_animals, 2022, doi:10.3390/ani12121560_

Round 1
Reviewer 1 Report
Authors aims to report the prevalence of ESBL producers in pig herds, to identify phenotypically and molecularly their antimicrobial resistance patterns and to investigate potential factors that could promote the development of AMR.
Minor comments/suggestion are highlighted in the attached .pdf file.
The authors satisfactorily made the comments and suggestions made by this reviewer, considering that the manuscript is in a position to continue with its process.
This is a topic of great interest, about a problem of importance worldwide and that gives us a vision of the dimensions, local in this case, of animal production and the development of ESBL bacteria and their potential role in public and animal health.

Author Response
We appreciate your suggestions.
Reviewer 2 Report
In attached file

Author Response
We appreciate your suggestions.
-Instead of the terms "genotyping" or "genotypic characterization", terms “antibacterial resistance genes” is more appropriate. "Genotyping" is a more complex process involving sequence determination and thus allows differences between strains to be found. and this statement should be corrected throughout the text.
Thank you for your comments. We corrected the term in all text.
-The materials and methods lack information on what genes were tested.
Thank you for your comments. We presented more data in an extra supplementary file S1
-In the results section, a simple table showing which genes and in what percentage were found could be useful. This is very important information and should be included in the manuscript (not in supplementary material)
Thank you for your comments. We added a new Table (Table 4)
-line 208-211 This section of the manuscript shows that 5 strains of Salmonella were not classified as ESCR, were they? However, the rest of the text shows that this was not the case.
Thank you very much for your comment. In the present study, Salmonella isolates were retrieved after non-selective cultivation according to ISO 6579-1:2017 and were subsequently phenotypically screened for ESBL production. As you noted, in lines 208-211 we show that we detected a total number of 5 Salmonella strains, among which, only one was classified as ESCR and ESBL producer (line 215). This finding is presented throughout the text and in Table 3.
-lines 210-216 Names of the bacteria in italics
Thank you for your comment. The names of bacteria were corrected.
-line 211 without “5/214”
Thank you for your comment. The Table was corrected.
-line 212 without “78/214”
Thank you for your comment. The Table was corrected.
-Table 3 without “%”. “Percentage “ is in in table headers
Thank you for your comment. The Table was corrected.
-Figure 3 What do numbers on y axis mean?
Thank you for your comment. The description of figure 3 provide all appropriate details.
-line 386 detected In this study?
Thank you very much for your comment. We clarified that we refer to the present study. of the revised text.
Reviewer 3 Report
The new version of the manuscript proposed by Tsekoras and colleagues was improved following my previous suggestion. Are still presents a few grammatical mistakes and typos.
I encourage the publication of this new versions, after the revision
Author Response
Thank you for your comments. We tried to improve the language style and correct the typos by
performing an extended revision of the original text.
This manuscript is a resubmission of an earlier submission. The following is a list of the peer review reports and author responses from that submission.
Round 1
Reviewer 1 Report
The paper of Tsekouras and colleagues aimed to detect antibiotic resistance in enteric pathogens from pigs.
The work is interesting and actual. The research is well performed and the results are clearly presented.
Here, my few comments:
-The language must be revised. Besides, several typos are present.
- please, include in the introduction and discussion the bacteria transmission between wild boar and domestic pigs. Similar results of your investigation were previously detected in wild boar (https://pubmed.ncbi.nlm.nih.gov/33498307/; https://pubmed.ncbi.nlm.nih.gov/32344604/)
- The inclusion and exclusion criteria from line 98 to line 115, in my opinion, could have better visualization in a table.
Reviewer 2 Report
Authors aims to investigate, for the first time in Greece, the prevalence of ESBL bacteria in swine populations and to correlate their occurrence with risk factors.
Specific suggestions/comments are highlighted in the attached .pdf file.
Can authors clarify the sample size number and strategy?
Did authors check for collinearity or association between the variables evaluated for the logistic regression?
The statement "This study reveals a positive correlation between herd size and AMR" is referred to a correlation analysis performed? if so, please incorporate it into the manuscript.
The topic is of high importance and would attract the interest of several readers, and the results of the study are of great importance, indicating the circulation of multiresistant bacteria in swines from Greece.
